# Involvement of Glucosamine 6 Phosphate Isomerase 2 (GNPDA2) Overproduction in β-Amyloid- and Tau P301L-Driven Pathomechanisms

**DOI:** 10.3390/biom14040394

**Published:** 2024-03-25

**Authors:** Mercedes Lachén-Montes, Paz Cartas-Cejudo, Adriana Cortés, Elena Anaya-Cubero, Erika Peral, Karina Ausín, Ramón Díaz-Peña, Joaquín Fernández-Irigoyen, Enrique Santamaría

**Affiliations:** Clinical Neuroproteomics Unit, Proteomics Platform, Navarrabiomed, Hospitalario Universitario de Navarra (HUN), IdiSNA, Navarra Institute for Health Research, Universidad Pública de Navarra (UPNA), Irunlarrea 3, 31008 Pamplona, Spain; mlachenm@navarra.es (M.L.-M.); pcartasc@navarra.es (P.C.-C.); acortesj@navarra.es (A.C.); elena.anaya.cubero@navarra.es (E.A.-C.); erika.peral.pintado@navarra.es (E.P.); karina.ausin.perez@navarra.es (K.A.); ramon.diaz.pena@navarra.es (R.D.-P.); jokfer@gmail.com (J.F.-I.)

**Keywords:** GNPDA2, neurodegeneration, olfaction, zebrafish

## Abstract

Alzheimer’s disease (AD) is a neurodegenerative olfactory disorder affecting millions of people worldwide. Alterations in the hexosamine- or glucose-related pathways have been described through AD progression. Specifically, an alteration in glucosamine 6 phosphate isomerase 2 (GNPDA2) protein levels has been observed in olfactory areas of AD subjects. However, the biological role of GNPDA2 in neurodegeneration remains unknown. Using mass spectrometry, multiple GNPDA2 interactors were identified in human nasal epithelial cells (NECs) mainly involved in intraciliary transport. Moreover, GNPDA2 overexpression induced an increment in NEC proliferation rates, accompanied by transcriptomic alterations in Type II interferon signaling or cellular stress responses. In contrast, the presence of beta-amyloid or mutated Tau-P301L in GNPDA2-overexpressing NECs induced a slowdown in the proliferative capacity in parallel with a disruption in protein processing. The proteomic characterization of Tau-P301L transgenic zebrafish embryos demonstrated that GNPDA2 overexpression interfered with collagen biosynthesis and RNA/protein processing, without inducing additional changes in axonal outgrowth defects or neuronal cell death. In humans, a significant increase in serum GNPDA2 levels was observed across multiple neurological proteinopathies (AD, Lewy body dementia, progressive supranuclear palsy, mixed dementia and amyotrophic lateral sclerosis) (*n* = 215). These data shed new light on GNPDA2-dependent mechanisms associated with the neurodegenerative process beyond the hexosamine route.

## 1. Introduction

Alzheimer’s disease (AD) represents the most frequent neurodegenerative disorder and has become a major public health problem worldwide [1,2]. Among other typical symptoms such as memory loss or disorientation, AD is characterized by an early severe olfaction loss in 90% of patients [3,4]. This early deficit has been associated with the appearance of neuropathological deposits such as amyloid plaques and neurofibrillary tangles [5]. The appearance of these neuropathological hallmarks is defined by the accumulation of beta-amyloid (Aβ) peptides and hyperphosphorylated Tau [6]. That is why therapeutic strategies are mostly Aβ and Tau-targeted; however, no definitive successful application has been described until now.

Dysregulation of brain energetics has been described in various neurodegenerative disorders, including AD [7,8]. Specifically, brain hypometabolism is a well-described hallmark in AD patients [9] and consistently, this event also occurs in animal AD models [10,11]. Although a decrease in glucose levels occurs progressively, the exact connection between glucose metabolism and AD is not yet understood. Together with an increased glycolytic ratio, a disruption of the hexosamine pathway and subsequent O-linked-N acetylglucosamine (O-GlcNAc) cycling has also been suggested as a link between altered glucose metabolism and brain defects [12]. In this context, previous reports have observed alterations in GNPDA2 protein levels in the olfactory bulb derived from AD and Parkinson’s disease subjects [13]. GNPDA2 is a metabolic enzyme that participates in the hexosamine pathway, one of the main nutrient-sensing pathways in several organisms, catalyzing the deamination of glucosamine-6-phosphate [14]. This chemical reaction results in two molecular events: (i) an increase in ammonium levels, a neurotoxic factor with great relevance in AD [15,16], and (ii) a decrease in UDP-GlcNAc levels, which consequently leads to O-GlcNAcylation decline. Interestingly, it has been previously suggested that a decline in APP and Tau O-GlcNAcylation might lead to the emergence of Aβ plaques and neurofibrillary tangles [17,18,19,20]. However, alterations in GNPDA2 gene/protein levels are mostly associated with obesity-related processes [21,22,23,24], and its potential involvement in neurodegeneration is unexplored. Our study aims to increase our knowledge about GNPDA2′s biology, clarifying its potential interactions in AD-related contexts. To achieve this purpose, we have combined RNA sequencing (RNA-seq) and quantitative proteomics with olfactory in vitro studies and in vivo experiments with a zebrafish transgenic line overexpressing the human P301L Tau mutant in a neuron-specific manner [25]. In addition, serum GNPDA2 levels have been monitored across multiple human neurological proteinopathies to extend our knowledge about hexosamine metabolism and neurodegeneration.

## 2. Materials and Methods

### 2.1. Materials

The following reagents and materials were used. Anti-GNPDA2 (ref. ab106363) and anti-Tau (phospho S396) (ab109390) were purchased from Abcam (Cambridge, UK). Anti-Znp1/Zn-1 antibody was obtained from the Developmental Studies Hybridoma Bank (Iowa, IA, USA). Anti-GAPDH (SAB2701826) was obtained from Sigma-Aldrich (Burlington, VT, USA) Anti-ERK antibody (9102) was purchased from Cell signaling (Danver, CO, USA), whereas anti-Tau K9JA was purchased from Dako (Glostrup, Denmark). Electrophoresis reagents were purchased from Biorad (Hercules, CA, USA) and trypsin from Promega (Madison, WI, USA).

### 2.2. Cell Culture

Human nasal epithelial cells (hNECs) (ABM. T9243) were cultured in DMEM (Gibco, Grand Island, NY, USA) supplemented with 10% FBS (Merck millipore, Burlington, MA, USA) and 1% penicillin/streptomycin (ABM) and grown in a 5% CO_2_ humidified atmosphere at 37 °C. For GNPDA2 overexpression, hNECs were seeded in 6-well plates (80% confluency). Briefly, one microgram of GNPDA2 and control GFP plasmid DNA were mixed with lipofectamine 3000 (Fisher, Wlatham, MA, USA) (1:2) in DMEM, followed by 15 min incubation at room temperature. Aβ 1–42 peptide (1 µM) (Sigma, Burlington, MA, USA) and Tau P301L plasmid (2.5 µg) (Addgene, Watertown, MA, USA) were added 24 h after. Cells were then incubated for 24 h and RNA was extracted using an RNeasy Mini Kit (Qiagen, Hilden, Germany) following the manufacturer’s instructions.

### 2.3. Immunoprecipitation

Human nasal epithelial cell (hNEC) protein extracts were homogenized on ice-cold immunoprecipitation (IP) buffer (50 mM Tris, 150 mM NaCl, 2 mM EDTA, 1% Triton, pH 7.4, 1 mM PMSF, 10 μg/mL aprotinin, 1 μg/mL leupeptin). Four hundred micrograms of protein were incubated overnight at 4 °C under rotary agitation with 2 µg of GNPDA2 antibody (Abcam ab106363) or ERK antibody (Cell signaling 9102) as a negative control. The immune complexes were incubated overnight with the addition of Sepharose beads (GE Healthcare, Chicago, IL, USA. 17-0618-01). Beads were washed once in IP buffer, IP/PBS (1:1) and PBS and collected via centrifugation at 3000 rpm for 5 min. Immunocomplexes were heated for 10 min at 100 °C and resolved on SDS-PAGE or resuspended in PBS. For mass-spectrometry analyses, immunoprecipitated protein extracts were diluted in Laemmli sample buffer and loaded into a 0.75 mm-thick polyacrylamide gel with a 4% stacking gel casted over a 12.5% resolving gel. The run was stopped as soon as the 3 mm of the front had entered the resolving gel, so that the whole interactome became concentrated in the stacking/resolving gel interface. Bands were stained with Coomassie Brilliant Blue and excised from the gel, and protein enzymatic cleavage was carried out with trypsin (Promega, Madison, WI, USA; 1:20, *w*/*w*) at 37 °C for 16 h, as previously described [26]. Purification and concentration of peptides were performed via C18 Zip Tip Solid Phase Extraction (Millipore, Burlington, VT, USA). Mass spectrometry analysis was performed using an EASY-1000 nanoLC system coupled to an Exploris 480 mass spectrometer (Thermo Fisher Scientific) as described below.

### 2.4. RNA Sequencing (RNA-Seq) and Data Analysis

Briefly, total RNA was extracted and purified using an RNeasy Mini Kit (Qiagen, Hilden, Germany) following the manufacturer’s instructions. Sequencing libraries were prepared by following the Illumina Stranded Total RNA Prep with Ribo-Zero Plus (Illumina Inc., San Diego, CA, USA) from 100 ng of total RNA, which was depleted by following the instructions. All libraries were run in a HiSeq1500 PE100 lane in rapid mode and pooled in equimolar amounts to a 10 nm final concentration. The library concentration was measured in Qubit 3.0 (Invitrogen, Waltham, MA, USA) and the library size was ensured by capillary electrophoresis in a fragment analyzer (AATI). The quality of the RNAseq results was initially assessed using FastQC v0.11.9 (http://www.bioinformatics.babraham.ac.uk/projects/fastqc/) and MultiQC v1.9 (http://multiqc.info/). The raw reads were trimmed, filtered for those with a Phred quality score of at least 25 and all adapters were removed with TrimGalore v0.5.0 (https://www.bioinformatics.brabraham.ac.uk/projects/trim_galore/). Trimmed reads were analyzed with SortMeRNA v2.1 software (https://bioinfo.lifl.fr/RNA/sortmerna/) [27] to delete the 18S and 28S rRNA to eliminate the rRNA residues that could remain undepleted by chemical treatment in the library preparation. Clean reads were aligned versus the Homo Sapiens reference genome (release GRCm38.p6/GCA_000001635.8, ftp://ftp.ensembl.org) using HISAT2 v2.2.1 (https://daehwankimlab.github.io/hisat2/) [28] with default parameters. Resulting alignment files were quality assessed with Qualimap2 (http://qualimap.bioinfo.cipf.es) [29] and sorted and indexed with Samtools software v1.18 [30]. After taking a read count on gene features with the FeatureCounts tool (http://subread.sourceforge.net) [31], a quantitative differential expression analysis between conditions was performed using DESeq2 [32], implemented in the R Bioconductor package, and read-count normalization was performed by following a negative binomial distribution model. In order to automate this process and facilitate all group combination analyses, the SARTools pipeline was used [33]. All resultant data were obtained as HTML files and CSV tables, including density count distribution analyses, pairwise scatter plots, cluster dendrograms, principal component analysis (PCoA) plots, size factor estimations, dispersion plots and MA and Volcano plots. The resulting CSV file, including raw counts, normalized counts, Fold-Change estimation and dispersion data, was annotated with additional data from the Biomart database (https://www.ensembl.org/biomart/martview/346d6d487e88676fd509a1b9a642edb2). In order to control the false discovery rate (FDR), the *p*-values were amended using Benjamini–Hochberg (BH) multiple testing corrections. Features showing corrected *p*-values below the 0.05 threshold and fold-change values of >1.5 or <0.5 were considered up- or down-regulated genes, respectively. (These websites were accessed on 14 February 2024).

### 2.5. Zebrafish Studies

The Zebrafish model characterization was performed using Biobide (https://biobide.com/; accessed on 14 February 2024). Briefly, adult zebrafish were housed and maintained in accordance with standard procedures. Fish were maintained in 3 L aquaria heated at 28.5 °C with about 20 fish per tank, and the water was continuously filtered. Water conditions were monitored and regulated conveniently. Fish were kept under a photoperiod of 14:10 h light/dark. Adults were fed with ground dry pellets and live food. Embryos were collected and placed in E3 media with ampicillin (100 μg/mL) and methylene blue (0.0001%) and provided immediately to the lab for injection. All experiments were performed according to European standards of animal welfare on animal used for scientific purposes (2010/63/EU), in addition to the national regulations for the care of experimental animals, and were approved as described in national regulations (RD 53/2013) by local and regional committees (internal codes: SUA-BBD-0003/19 (WT production and maintenance; authorization code: PRO-AE-SS-158) and SUA-BBD-0005/19 (Transgenic ZF production, no pathology associated; authorization code: PRO-AE-SS-151)). Plasmid-containing GNPDA2 cDNA (clone OHu09718 in pBluescript II KS(+)) was commercially obtained from GenScript. Plasmid (5 μg) was linearized with SalI following the manufacturer’s protocol. Once digestion was complete (verified on an agarose gel), the sample was treated with proteinase K (100 μg/mL) and SDS (0.5%) and then DNA was purified with phenol/chloroform. GNPDA2 mRNA was synthetized from this linearized plasmid using an mMessage mMachine T7 Ultra kit (Ambion, Austin, TX, USA) following the manufacturer’s instructions. The mRNA control for Cypridine luciferase (cLuc) was also synthetized as described above using the cLuc control template provided in a HiScribe T7 ARCA mRNA kit (New England BioLabs, Ipswich, MA, USA). After checking that synthesis was successful (agarose gel for RNA detection), poly A-tail was added (Poly A tailing Kit, Ambion) to both mRNAs following the manufacturer’s instructions. Finally, mRNAs were purified via Qiagen columns also following the manufacturer’s protocol, and the concentration of mRNAs obtained was determined via absorbance and densitometry (ImageJ software v1.54f) (Appendix A).

### 2.6. RNA Injection, Evaluation of Toxicity and Determination of GNPDA2 Protein Expression

Wild-type AB embryos at the one-cell stage were placed in a premade 2% agarose plate that had series of wedge-shaped grooves, where embryos were aligned and immobilized. mRNA solutions (mRNA + 100 mM KCl + 0.5 mg/mL Phenol Red) were loaded into the injection capillary and delivered directly into the cell. A volume of 1–2 nL of the two mRNAs was injected at three different doses (based on the final RNA concentration determined by the absorbance) and around 100 embryos were injected per mRNA/concentration. Once injected, embryos were transferred into a Petri dish with E3 embryo media + 10 mM HEPES and incubated at 28.5 °C until the end of the experiment (48 hpf). Embryos not injected for the same batch/batches used for injection were also kept for comparison. The presence of toxicity manifestations (including mortality and morphological alterations) was evaluated at 24 and 48 hpf under a stereoscope in all the experimental groups (Appendix A). Once the analysis at 48 hpf was completed, 35 embryos were collected and deyolked by pipetting 5 times with a P200 pipette tip in ice-cold PBS. After washing, embryos were lysed in Rippa buffer (without SDS) and frozen at −80 °C. Cell extracts were then fractionated electrophoretically, transferred into nitrocellulose filters and subjected to an immunoblot analysis with an antibody that specifically recognizes GNPDA2. Endogenous Gapdh was also analyzed as a loading control. Immunoreactive bands were developed using an enhanced chemiluminescence system (LI-COR C-Digit Blot Scanner) (Appendix A). One RNA concentration (the highest tested that did not induce unspecific toxicity at which protein expression was confirmed) was selected for next steps.

### 2.7. RNA Injection and Evaluation of the Phenotype Associated with TAU P301L Overexpression

For the overexpression experiments, GNPDA2 and control Luciferase mRNAs were injected into one-cell-stage embryos from the ZF TAUP301L transgenic line at the selected concentration (384 ng/μL). Around 200 embryos were injected. A group of not injected embryos from the same transgenic line were retained for further comparisons. At 24 h post fertilization (hpf), dead and malformed embryos were removed and the DsRed expression (indicative of Tau expression) was determined under a fluorescence stereoscope. Embryos expressing high levels of DsRed, as well as DsRed-negative embryos (control siblings), were selected for further analysis. DsRed-negative embryos were used to determine the possible effect of RNAs injected in the absence of any neuronal damage and to verify the neurotoxicity caused by TAUP301L overexpression.

Concerning GNPDA2 overexpression, around 100 embryos were injected with a volume of 1–2 nL of mRNA. Once injected, embryos were transferred into a Petri dish with E3 embryo media + 10 mM HEPES and incubated at 28.5 °C until the end of the experiment (48 hpf). The experimental groups are defined in Appendix A.

### 2.8. Axonal Motoneuron Extension

For whole-mount immunohistochemistry, embryos from the different experimental groups were fixed at around 30 hpf in 4% (wt/vol) paraformaldehyde (PFA) in PBS overnight at 4 °C. Around 20 embryos were used per experimental group. Once fixed, embryos were dehydrated in 100% methanol and kept at −20 °C until immunohistochemistry was performed. At this time, before incubation with the primary antibody, embryos were rehydrated, permeabilized with 10 μg/mL proteinase K and blocked in 5% normal serum, 4 mg/mL BSA and 0.5% Triton x-100 in PBS. Incubation with two primary (anti-znp1 1:500; anti-zn1 1:200, which label the synaptic protein synaptotagmin in extending axons of all primary motoneurons) and secondary (goat anti-mouse Alexa Fluor 488, dilution 1:500) antibodies was performed overnight at 4 °C. Labeled embryos were imaged on a fluorescent stereomicroscope (Leica M20FA) and pictures of the area of interest (the first 4 stained motoneuron projections before the end of the yolk extension) were obtained with a digital camera (KI5, Leica, Wetzlar, Germany). Motoneuron axon length measurements were performed using ImageJ software v1.54f (National Institutes of Health, Bethesda, MD, USA). A statistical analysis (one-way ANOVA plus Tukey’s multiple comparison test) was conducted to determine if expression of GNPDA2 affected motoneuron axonal extension when compared with not-injected or embryos injected with RNA control, both in the presence (DsRed-positive) and absence (DsRed-negative) of Tau P301L overexpression.

### 2.9. Detection of Neuronal Death

Neuronal death was studied via a TUNEL assay in all experimental groups at 48 hpf (around 20 embryos per group). At the indicated stage, embryos were fixed, dehydrated, rehydrated and permeabilized as previously described, and the TUNEL reaction (ApopTag Apoptosis Detection Kit, Millipore, Burlington, VT, USA) was carried out following manufacturer’s instructions. Embryos were imaged using a fluorescent stereomicroscope (Leica M20FA), and the number of positive neuronal nuclei for each experimental group was counted along the spinal cord area from the end of the most bulging part of the yolk to the tail tip. Representative pictures were taken with the digital camera previously described. A statistical analysis (one-way ANOVA plus Tukey’s multiple comparison test) was conducted to compare the number of apoptotic cells between all the experimental groups to determine if GNPDA2 expression affected neuronal apoptosis both in the absence and presence of Tau P301L overexpression.

### 2.10. Detection of TAU Phosphorylation

A group of 30 DsRed-positive embryos from each of the experiments were collected at 48 hpf, deyolked and lysed. The expression of phosphorylated TAU protein specifically at residue Ser396 was evaluated via Western blotting (as described in the same task) using an anti-PhosphoTAU (Ser396) antibody (Abcam). Same samples were developed in parallel with an anti-TAU antibody that recognizes both phosphorylated and non-phosphorylated TAU and with an anti-GNPDA2 antibody to confirm GNPDA2 expression. Immunoreactive bands were developed using an enhanced chemiluminescence system (LI-COR C-Digit Blot Scanner) and quantified later via densitometry using Image Studio software v1.54f. As a loading control, Ponceau S red staining of the membrane fragments used for the development of PhosphoTAU and total TAU was conducted; pictures were taken (G:BOX, Syngene, Cambridge, UK) and densitometry of the corresponding lines was performed using ImageJ software v1.54f.

### 2.11. Embryo Collection and Sample Preparation for Proteomic Analysis

Embryos from each of the experimental conditions (around 50 embryos per group) were collected at 48 hpf, deyolked, washed, frozen in liquid nitrogen and stored at −80 °C until further processing for the mass spectrometry experiment. Briefly, embryos were lysed with a buffer containing 7 M urea (Sigma, Burlington, MA, USA), 2 M tiourea (Sigma) and 50 mM ABC. Protein extracts were then quantified using a Bradford assay and 20 µg was utilized for the proteomic analysis. The reduction step was performed by adding dithiothreitol (DTT) (Sigma) to a final concentration of 10 mM and incubating at RT for 30 min. Subsequent alkylation with 30 mM (final concentration) iodoacetamide (IAA) (Sigma) was performed for 30 min in the dark at RT. Then, an additional reaction step was performed with 30 mM DTT (final concentration) for 30 min. This mixture was diluted to 0.9M urea using Miliq-water. The digestion step was performed using trypsin (Promega, Madison, WI, USA) at 1:50 *w*/*w* (enzyme/protein) and samples were incubated at 37 °C for 16 h. Digestion was quenched by acidification with acetic acid. The final step before mass spectrometry was the Vacuum Manifold platform. Then, samples were dried via a vacuum centrifuge and resuspended in 10 µL of 2% acetonitrile, 0.1% formic acid and 98% miliQ water.

### 2.12. Mass Spectrometry

Dried peptide samples were reconstituted with 2% ACN-0.1% FA (acetonitrile/formic acid) and quantified via a NanoDropTM spectrophotometer (ThermoFisher Sci., Waltham, MA, USA) prior to LC-MS/MS. Samples were analyzed using an EASY-1000 nanoLC system coupled to an EZ-Exploris 480 mass spectrometer (Thermo Fisher Sci.). Peptides were resolved using a C18 Aurora column (75 µm × 25 cm, 1.6 µm particles; IonOpticks) at a flow rate of 300 nL/min using a 60 min gradient (50 °C): 2–5% B in 1 min, 5–20% B in 48 min, 20–32% B in 12 min, and 32–95% B in 1 min (A = FA, 0.1%; B = 100% ACN: 0.1% FA). Sample data were acquired in data-independent acquisition (DIA) mode with a full MS scan (scan range: 400 to 900 *m*/*z*; resolution: 60,000; maximum injection time: 22 ms; normalized AGC target: 300%) and 24 periodical MS/MS segments applying 20 Th isolation windows (0.5 Th overlap; resolution: 15,000; maximum injection time: 22 ms; normalized AGC target: 100%). Peptides were fragmented using a normalized HCD collision energy of 30%.

### 2.13. Data Analysis

The resulting mass spectrometry data files were analyzed using Spectronaut 17.1 (Biognosys Schlieren, Schlieren, Switzerland) via a direct DIA analysis (dDIA) against a Danio rerio Swissprot (isoforms) database using the default settings and filtering the precursors and protein groups using a 1% q-value. The enzyme was set to trypsin in a specific mode (maximum of two missed cleavages). Carbamidomethyl (C) was set as fixed modifications. Oxidation (M), Acetyl (Protein N-term), Deamidation (N), and Gln -> pyro-Glu were set as variable modifications (3 maximum modifications per peptide).

### 2.14. Bioinformatic Analysis

The quantitative data obtained in Spectronaut software v18 were analyzed using Perseus software (version 1.6.15.0) [34] in order to perform a statistical analysis and visualize the obtained data. Statistical significance was set at a *p*-value lower than 0.05 in all cases, and a 1% peptide FDR threshold was considered. In addition, proteins were considered significantly differentially expressed when their absolute fold change was below 0.77 (down-regulated proteins) and above 1.3 (up-regulated proteins) on a linear scale. The association of the differentially expressed genes and proteins with specifically dysregulated regulatory/metabolic networks was analyzed using different bioinformatic tools (Metascape, Biogrid, QIAGEN’s Ingenuity Pathway Analysis v107193442 (IPA; QIAGEN Redwood City, CA, USA), SynGO). Metascape was used to extract the biological information associated with proteome functionality [35]. Biogrid was used to analyze potential APP and Tau interactors. On the other hand, IPA software v1.2 calculates significant values (*p*-values) between each biological or molecular event and the imported molecules based on Fisher’s exact test (*p* ≤ 0.05). The IPA comparison analysis hierarchically considers and reports the signaling pathway rank according to the calculated *p*-value.

### 2.15. Enzyme-Linked Immunosorbent Assay

GNPDA2 concentrations were measured using enzyme-linked immunosorbent assay (ELISA) kits according to the manufacturer’s instructions (MBS9341798—Mybiosource, San Diego, CA, USA). The data were analyzed using Graphpad Prism Software v8 and a Mann–Whitney U test was performed to make group comparisons. A *p*-value less than 0.05 was considered statistically significant. Serum samples included are indicated in Table 1 (Appendix A).

## 3. Results

### 3.1. GNPDA2 Interactome Is Partially Associated with Intraciliary Transport in NECs

The physiological and molecular role of GNPDA2 in olfactory contexts is not yet completely understood. This is why we believed that the elucidation of the GNPDA2 interactome in a human nasal epithelial cell line could help us to decipher its direct association with specific cellular processes. Hence, GNPDA2 was immunoprecipitated from hNECs as well as an irrelevant antibody (ERK) which was included as a negative control to detect non-specific associated proteins. Additionally, immunoprecipitation was also performed without using any specific antibody as a second negative control in the process. First, GNPDA2 immunoprecipitation was confirmed using Western blotting (Figure 1A). A subsequent mass spectrometry analysis was performed to identify GNPDA2-associated proteins (Figure 1A and Appendix A). Data were curated by excluding non-specific associated proteins detected in both ERK and no antibody immunoprecipitations (Appendix A). After restrictive curation, 93 proteins were considered as co-immunoprecipitating with GNPDA2. The GNPDA2 interactome was functionally analyzed using Metascape. As shown in Figure 1A, multiple functional categories were significantly enriched (Appendix A). Intraciliary transport (GO:0035735), metabolism of RNA (R-HAS-8953854) and the amino-acid metabolic process (GO:0009081) were represented by 17 (PCM1, CEP131, IFT81, AP2M1, IFT172, LRRC7, ATP2B4, TUBB2A, NDC80, SPAG5, TPPP, PPP2R3C, PARD3, SIRT5, TGS1, CCDC96), 16 (POLR2L, RPS28, WDR46, CNOT8, ZC3H11A, SMG7, SMNDC1, SMG6,RPL26L1, TGS1, LAMTOR3, ATP6V1G1, AP2M1, MBP, TUBB2A, YES1) and 3 proteins (ALDH6A1, PCCA, HIBCH), respectively. Cellular component analysis revealed that GNPDA2-specific interactors were mainly defined as membrane-related and nuclear, indicating a scattered cell localization (Figure 1B). Finally, we observed that fourteen GNPDA2 interactors (APBB1, ATP6V1G1, TPPP, GFAP, GRIN2A, NFKBIB, UGCG, C1orf174, ATPAF2, IFT81, FRMD8, SIRT5, RPL26L1, LRRC42) had previously been described as APP interactors, whereas another two of them (F2, TUBB2A) were experimentally demonstrated Tau interactors (Figure 1C).

### 3.2. Dissimilar Transcriptomic Variability in GNPDA2-Overexpressing NECs upon Neuropathological Insult

Due to the increased GNPDA2 protein levels, which have previously been observed in neuropathological olfactory-related contexts [13,36], we explored the potential molecular events linked to GNPDA2 overexpression in NECs in the presence of neuropathological insults (beta amyloid (Aβ) and the mutated form of h.Tau P301L). The experimental design is shown in Figure 2A. First, we observed that GNPDA2 overexpression had a positive effect on cell proliferation, accompanied by transcriptomic changes in 172 genes (Figure 2B). On the contrary, the presence of both Aβ and Tau P301L triggered a reduction in the NEC proliferation capacity (Figure 2B). However, Aβ-induced transcriptomic modulation was more severe than that observed in GNPDA2-overexpressing NECs overproducing h.Tau P301L (1879 and 229 differentially expressed genes, respectively; *p*-value adjusted <0.01), suggesting a synergic effect in the process of neurodegeneration (Figure 2B and Appendix A). Only 11 were commonly deregulated in the three experimental conditions (Figure 2C). A functional analysis demonstrated that Aβ-impaired biological pathways related to extracellular matrix organization, actin filament-based processes, regulation of cell projection organization and the cellular response to growth factor stimulus, among others, suggesting a negative impact in cell structure and response against external signaling (Figure 2D and Appendix A). On the other hand, the concomitant expression of GNPDA2 and h.Tau P301L seemed to interfere specifically with protein processing in the endoplasmic reticulum, PID integrin development, and membrane-bounded organelle assembly, among others (Appendix A). To characterize in detail the transcriptomic alterations in the three experimental conditions, transcriptome-scale interaction network analysis was performed. As shown in Figure 3, GNPDA2 overexpression mainly induced downregulation in protein components associated with IFN alpha interactome. On the other hand, the massive molecular disruption accompanying GNPDA2 overproduction in the presence of Aβ mainly revealed alterations in AKT signaling dynamics whereas Tau P301L expression impacted on other survival kinases such as ERK and P38 MAPK (Figure 3).

### 3.3. Effects of Human GNPDA2 Overexpression in h.Tau P301L Transgenic Zebrafish Embryos

To elucidate the role of GNPDA2 in an in vivo neurodegenerative context, we explored the concomitant expression of human Tau P301L and GNPDA2 in a transgenic zebrafish line, combining functional assays with quantitative proteomics. The experimental design is detailed in Figure 4A. As indicated, we first investigated the relationship between GNPDA2 overexpression and neuronal toxicity. Impairment of axonal outgrowth is considered a hallmark of tauopathy [25,37] and is observed in Tau P301L embryos. Hence, to examine abnormalities in axonal growth, Tau P301L transgenic embryos overexpressing GNPDA2 were immunostained with anti-zpn1/zn-1znp1 antibodies, labelling the synaptic protein synaptotagmin in the extending axons of all primary motoneurons. Not injected Tau− and Tau+ embryos were analyzed as the animal model control, whereas cypridine luciferase (cLUC) Tau− and Tau+ embryos were considered transfection controls. As shown in Figure 4B, a statistically significant decrease in the axonal length was associated with h.Tau P301L overexpression. However, human GNPDA2 expression did not prevent or increase this effect (Figure 4B). In addition, there was a clear increase in the number of apoptotic cells in the spinal cord of Tau P301L embryos, without detecting neither increase nor decrease in the number of these cells due to GNPDA2 overproduction (Figure 4C). Alterations in the phosphorylated Tau (ser396) levels were analyzed via Western blotting at 48 h post fertilization (hpf). An increase in the amount of phosphorylated Tau was observed in embryos expressing GNPDA2, probably due to a slight observed increase in total Tau levels (Figure 4D). These data indicate that overexpression of human GNPDA2 together with the h.Tau P301L mutation in zebrafish neurons does not modify the neuronal toxicity but slightly interferes with Tau levels.

### 3.4. Proteomic Analysis Revealed Widespread Alterations in GNPDA2-Overexpressing h.Tau P301L Zebrafish Embryos

To examine the molecular alterations occurring in the development of the zebrafish expressing both human Tau P301L and GNPDA2 proteins, a nanoLC-MS/MS workflow based on an EASY-1000 nanoLC system coupled to an Exploris 480 mass spectrometer with a data-independent acquisition approach was applied to zebrafish specimens derived from all experimental groups of the study (Figure 4A). Among 1300 quantified proteins with at least two peptides, a differential expression analysis was performed between the above-mentioned experimental groups. First, we initially explored the molecular impact of Tau P301L mutations in the zebrafish model, establishing a differential between not-injected Tau− embryos and not-injected Tau+ embryos. Forty-five statistically differentially expressed proteins (DEPs) were specifically altered due to the overexpression of Tau (Figure 5A), pointing out alterations in protein-translation-related pathways, apoptosis and lipid metabolism, among others (Figure 5B). Moreover, an additional analysis indicated that the overexpression of Tau P301L in zebrafish specimens had an impact in a small percentage of previously described Tau interactors together with synapse-related proteins (Figure 5C,D). To analyze the potential effect of GNPDA2 overexpression on Tau P301L zebrafish embryos, an ANOVA test was performed, comparing protein levels between the following experimental groups: not injected Tau−, not injected Tau+, cLUC Tau+ and GNPDA2 Tau+. Among the 823 DEPs, we initially observed 12 protein subsets modulated among the different experimental groups (Figure 6A). Interestingly, we found that GNPDA2 overexpression specifically modulated two specific clusters targeting the citric acid cycle and respiratory electron transport chain, carbon metabolism and cell-structur-related pathways such as actin polymerization, collagen biosynthesis and axis elongation (Figure 6B; clusters 1 and 2). In addition, the parallel expression of human Tau P301L and GNPDA2 had a greater impact in a subgroup of proteins related to the regulation of axonogenesis (Figure 6B; cluster 3). Moreover, human GNPDA2 expression was able to revert the pathological effect of 69 proteins altered in the presence of Tau P301L. Fifty of them (mainly involved in endocytosis and LCAM1 interactions) had diminished levels after GNPDA2 overexpression, whereas levels of nineteen proteins (most of them ribosomal proteins) were increased (Figure 6B; clusters 4 and 5).

### 3.5. Increased Serum GNPDA2 Protein Levels across Neurodegenerative Proteinopathies

Due to GNPDA2 interfering with multiple metabolic routes in different neurodegenerative contexts, we decided to evaluate GNPDA2 protein levels in serum across different neurodegenerative diseases such as AD, Lewy body dementia (LBD), progressive supranuclear palsy (PSP), frontotemporal dementia (DFT) and amyotrophic lateral sclerosis (ALS) (*n* = 187) (Figure 7A). We found that GNPDA2 serum levels were commonly increased across all indicated disorders with respect to the control population, and sex-independent in LBD, PSP, DFT and ALS patients and in control subjects (Figure 7B, Appendix A). On the contrary, as shown in Appendix A, GNPDA2 levels were higher in women than in men in AD patients. These findings partially suggest that GNPDA2 may be considered as a widespread marker of neurodegeneration.

## 4. Discussion

The lack of understanding of the disease mechanisms that underlie a vast amount of neurodegenerative processes hinders the discovery of early disease biomarkers and the development of efficacious therapeutic approaches. Concerning AD, amyloid plaques and intracellular neurofibrillary tangles represent the two main neuropathological hallmarks present in the brain of AD subjects [6]. Although there is an extensive debate concerning the actual role of Aβ and Tau in the AD neurodegenerative process, new currents of thought suggest that the pathology may come from a lack of function of the soluble form of these two molecules [38]. Among the numerous mechanisms that connect the neurotoxic Aβ and Tau are oxidation of proteins, lipid peroxidation and brain hypometabolism. In fact, the glucose decline puts the brain at risk concerning cognitive functions, and cerebral metabolic rates of glucose represent an early observation in AD subjects [9]. In this context, GNPDA2 is a metabolic enzyme that actively participates in the hexosamine pathway [14]. In this study, we have partially defined the GNPDA2 interactome using immunoprecipitation in a human nasal epithelial cell line coupled with mass spectrometry. To our knowledge, this is the first study that experimentally describes the GNPDA2 interactome and we found that most GNPDA2 interactors participate in intraciliary transport. The nasal cavity is covered by a pseudostratified ciliated epithelium that has a front-line role in respiratory defense [39], and the physical interaction between GNPDA2 and ciliary and membrane proteins such as PCM1, CEP131, IFT81, AP2M1, IFT172, LRRC7, ATP2B4 or TUBB2A suggests a potential new function of GNPDA2 in a physiological context. In line with these results, we found that the overexpression of GNPDA2 in the presence of Aβ promoted alterations in cell projection organization. Accumulation of Aβ at presynaptic terminals has been shown to trigger primary cilia dysfunction and, consequently, neurite dystrophy [40]. On the other hand, we found that 14 GNPDA2 interactors had previously been described as APP interactors. Among all these protein partners, N-methyl-D-aspartate receptor GRIN2A expression levels were additionally altered in hNECs after GNPDA2 overexpression and Aβ treatment. GRIN2A belongs to the family of ionotropic glutamate N-methyl-D-aspartate receptors (NMDARs), with essential roles in synaptic plasticity [41]. In fact, previous studies have demonstrated that GRIN2A knockdown accelerates memory and cognitive deficits in mice [42]. Hence, although more studies are needed in other neurological contexts, these data suggest a potential synergic effect between GNPDA2 and GRIN2A in AD progression.

Based on the involvement of olfactory dysfunction in AD, we used hNECs to examine whether GNPDA2 overexpression might be responsible for enhancing the neurodegenerative process. The RNA-seq analysis revealed a slight modulation of hNECs transcriptome when overexpressing GNPDA2, specially exacerbated when combined with Aβ. Specifically, our data pointed out that GNPDA2 overexpression induces a shutdown of several ubiquitin-related genes (HERC6, TRIM22, IG15, PLAAT4) potentially governed by IFN alpha. Interestingly, previous reports have demonstrated that increasing O-GlcNAcylation levels promote ubiquitination [43,44]. In this context, many neurodegenerative diseases are associated with the formation of ubiquitin-conjugated protein aggregates, including Aβ and Tau aggregates [45]. Hence, GNPDA2 overexpression appears to be a driver of the neurodegenerative process leading to alterations in both O-GlcNAcylation and ubiquitination processes. On the other hand, complementing signaling mapping, AKT appeared as the main node when analyzing the differential transcriptome after overexpressing GNPDA2 and Aβ. It is well known that AKT plays a crucial role in AD progression [46], and AKT hyperactivation has also been demonstrated in olfactory regions in AD and PD subjects [13,36]. On the contrary, after analyzing the upstream signaling interactions of the differentially expressed transcriptomes after the concomitant expression of GNPDA2 and Tau P301L, upstream regulators such as ERK1/2 and p38 MAPK were reported to be highly interconnected with downstream altered genes. Previous reports in nasal epithelial cells have described the activation of ERK and p38 MAPK as a damage response [47,48]. On the other hand, these two pro-survival kinases have also been linked with AD-related processes such as APP processing and Tau phosphorylation [49,50,51,52,53]. In this regard, activation of ERK was evidenced in the OB of AD subjects, whereas early activation of p38 MAPK was reported in initial AD stages [36]. Thus, our data suggest that GNPDA2 overexpression in hNECs leads to stress responses in the cell accompanied by alterations in survival-related kinases.

Animal models are critical to increase the understanding of AD and to evaluate potential new therapeutic options [54]. Several transgenic models have been generated round the biology of Tau aiming to define the molecular mechanisms underlying neurodegenerative processes [55,56,57,58]. In this study, to determine whether GNPDA2 signaling could enhance the neurodegenerative process in vivo, we took advantage of the zebrafish Tau P301L line [25]. Corroborating previous studies that have previously linked ribosomal dysfunction to tauopathies [59,60], we found that Tau P301L zebrafish embryos presented alterations in various ribonucleoproteins. In addition, Tau P301L triggered apoptotic-related events, impacting the antiapoptotic BH3 family [61]. Saying that, this is the first study revealing additional molecular alterations derived from Tau P301L mutant overexpression in zebrafish. Conversely, although GNPDA2 overexpression did not exacerbate or reduce axonal motoneuronal extension or neuronal death, it triggered significant proteome alterations at 48 hpf in this zebrafish model, involving axonogenesis-related proteins (dpysl3, dpysl2b, dbn1, vat1, cnmd, ndr2). In this sense, little is known concerning the role of GNPDA2 besides its role in the hexosamine pathway [14]. In fact, other studies have mainly linked its activity with obesity-related processes. To our knowledge, this is the first study linking GNPDA2-specific modulation with other cell processes such as axonogenesis, the citric acid cycle, respiratory electron transport, carbon metabolism, RNA splicing and the regulation of actin filament polymerization or mitotic anaphase.

Conversely, ammonia, the product of GNPDA2 activity, has already been considered as a neurotoxic factor, with severe negative effects on the central nervous system [16]. Excessive levels of ammonia have also been shown to provoke axonogenesis impairment and irreversible brain damage in neonates [62], and increased levels of ammonia have been detected in the brain and blood of AD patients [15]. This has shown to trigger inflammatory responses and the release of cytokines and inflammatory proteins by microglia, astrocytes and neurons via apoptosis and neuronal degeneration [63]. Studies have suggested the over-activity of adenosin-3-monosphosphate (AMP) deaminase as the main source of elevated ammonia levels [64]. However, our findings have shown increased levels of GNPDA2 not only in AD patients but also in a wide variety of neurodegenerative disorders such as Lewy body dementia, progressive supranuclear palsy, mixed dementia and amyotrophic lateral sclerosis, strongly suggesting GNPDA2 as another source of excessive ammonia. In this scenario, it is important to note that neurological disorders show strong sex differences in both disease incidence and manifestation [65,66], and it is now well established that understanding sex differences in the pathogenesis of AD and other neurodegenerative disorders is key to improving the development of effective diagnostics and therapeutic treatments. Importantly, significant GNPDA2 alterations were found to be sex-independent in LBD, PSP, DFT and ALS patients. No differences were found in control population also. Interestingly, GNPDA2 proteins levels were higher in AD women compared to men. There is growing evidence supporting sex differences in the incidence of AD [67,68], and, in fact, sex differences have been found in glucose metabolism in AD patients [69], supporting a potential involvement of GNPDA2 in this neurological context. To conclude, concerning the urgency that exists regarding the discovery of neurodegenerative drivers, our findings provide new molecular insights regarding the imbalance in the hexosamine metabolic intermediates during neurodegenerative process.

## Figures and Tables

**Figure 1 biomolecules-14-00394-f001:**
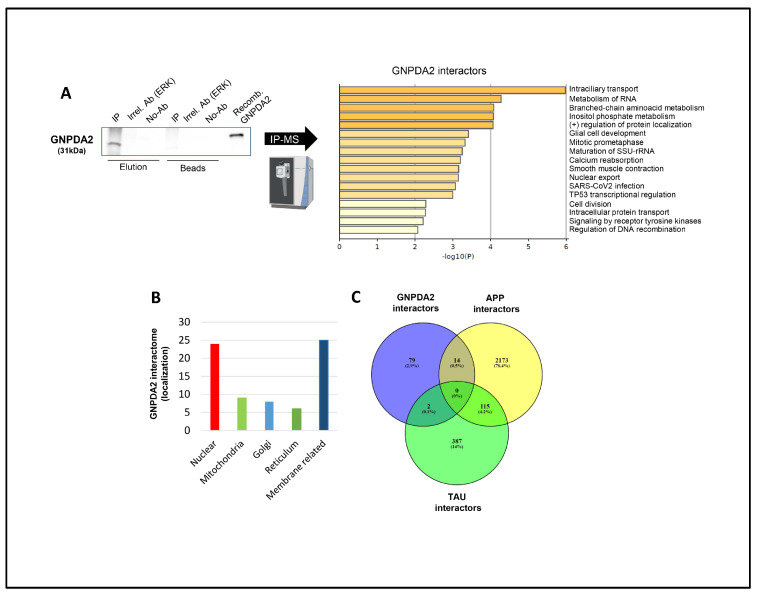
Identification of GNPDA2 molecular interactors. (**A**) An immunoprecipitation (IP) assay was performed in protein extracts from a human olfactory cell line. ERK antibody and recombinant GNPDA2 (6xHis-tagged) were used as assay controls (left). Metascape analysis indicating the biological pathways governed by GNPDA2 interactors (right). (**B**) GO analysis showing cellular localization of specific GNPDA2 interactors. (**C**) Venn diagram showing common GNPDA2, APP and Tau interactors.

**Figure 2 biomolecules-14-00394-f002:**
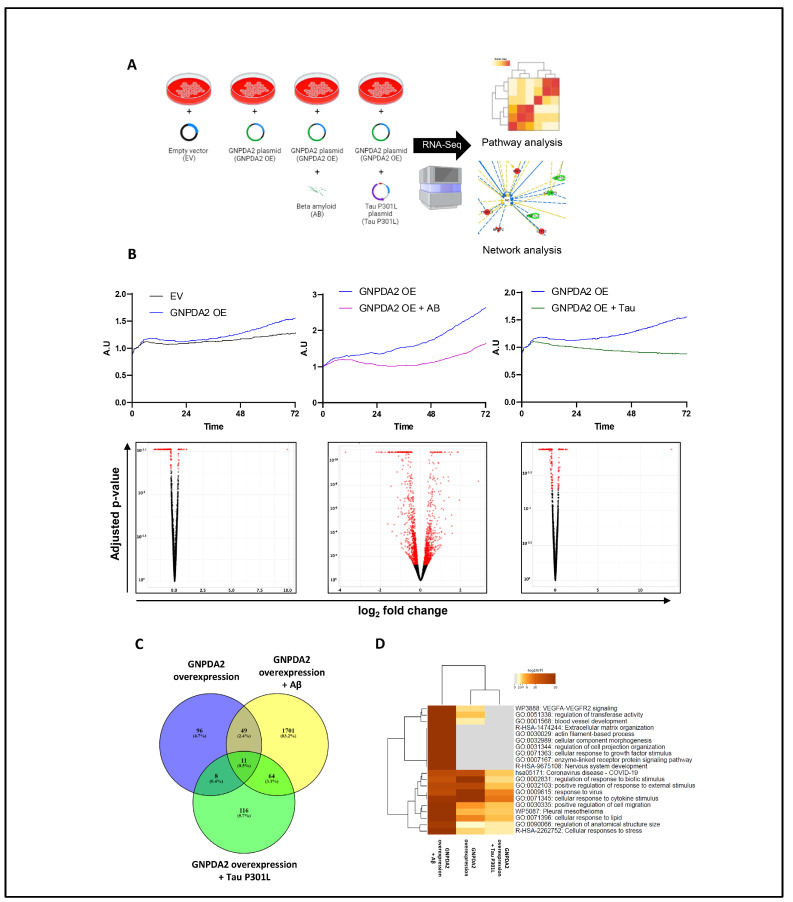
RNA-seq analysis in hNECs. (**A**) Brief description of experimental design. (**B**) Cell proliferation monitoring (via an xCELLigence analysis, Agilent, SC, USA) and the corresponding Volcano plots of differentially expressed genes (in red) across biological conditions: effect of GNPDA2 overexpression in hNECs, effect of Aβ treatment after GNPDA2 overexpression in hNECs and effect of Tau P301L treatment after GNPDA2 overexpression in hNECs. (**C**) Overlap across differential RNA-seq datasets. (**D**) Top 20 functional analyses across the three biological conditions generated by Metascape.

**Figure 3 biomolecules-14-00394-f003:**
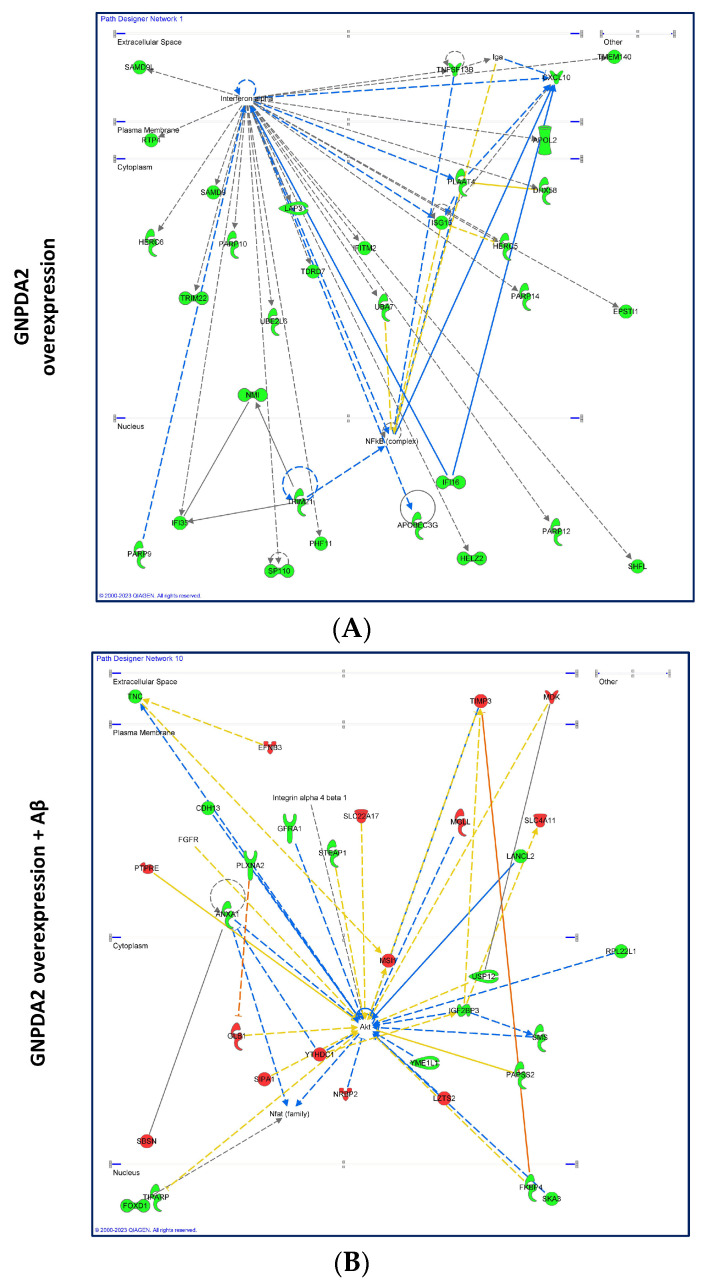
Gene interactome maps for differentially expressed genes in hNECs. Representation of the interactions between differentially expressed genes after GNPDA2 overexpression (**A**), GNPDA2 overexpression and Aβ treatment (**B**) and GNPDA2 and Tau overexpression (**C**). The color of each gene represents its up (red) and down (green) expression found in the RNAseq experiment. White nodes represent targets potentially responsible for the deregulation of the surrounding genes.

**Figure 4 biomolecules-14-00394-f004:**
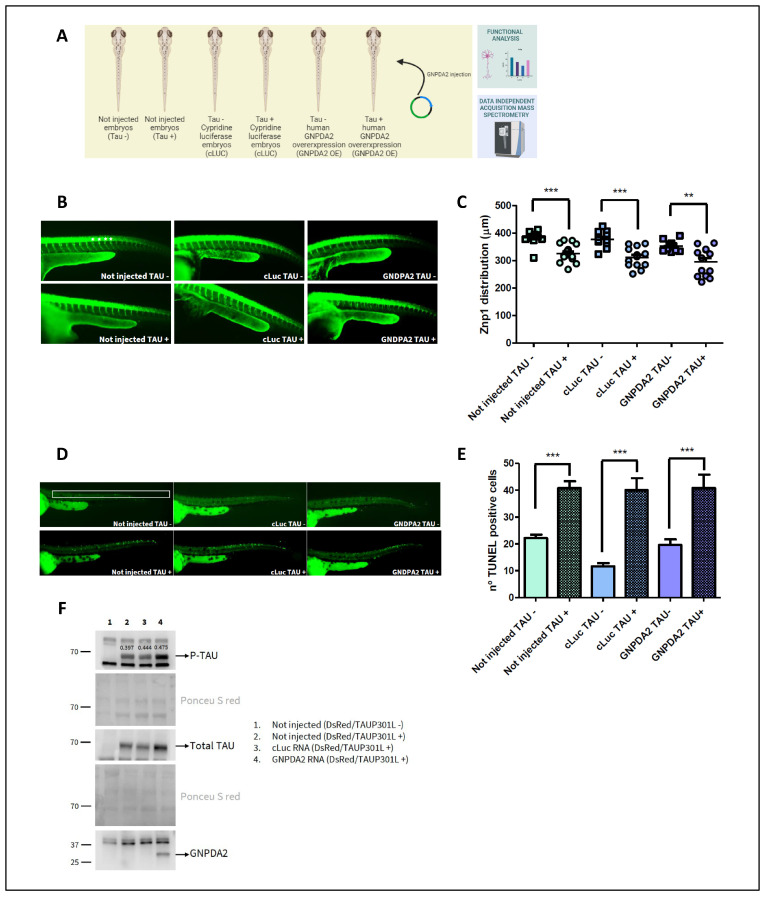
Study of GNPDA2 overexpression in zf Tau P301L transgenic embryos. (**A**) Experimental design used in the zebrafish model. (**B**) Representative pictures of 30 hpf not-injected embryos or embryos injected with cLuc and GNPDA2 mRNAs (both DsRed/mutant TAU-positive and -negative embryos) after whole-mount immunostaining with znp1/zn-1 antibodies. White asterisks indicate the position of the four axonal extensions quantified in not injected Tau− pictures. (**C**) Graph shows the mean ± SEM of the total axon length of the first four caudal primary motoneurons anterior to the end of the yolk extension of the indicated experimental groups. ** *p* < 0.01; *** *p* < 0.001. (**D**) Representative pictures of 48 hpf not-injected embryos or embryos injected with cLuc and GNPDA2 RNAs (both DsRed/mutant TAU-positive and -negative embryos) after TUNEL staining. The spinal cord area, where the number of TUNEL-positive cells was quantified, is highlighted in not injected Tau− pictures. (**E**) Graph shows the mean ± SEM of the number of TUNEL-positive neurons present along the spinal cord area of the indicated experimental groups. *** *p* < 0.001. (**F**) Western blot developed with anti-Tau (Phospho S396), anti-Tau (total) and anti-GNPDA2 antibodies of 48 hpf embryos from the indicated experimental groups. Ponceau S red staining corresponding to fragments used for developing pTau and total Tau are shown below each antibody-specific staining. Numbers correspond to the optical density quantification (arbitrary units) of Ser396-phosphorylated Tau with respect to the total Tau levels. Specimens in the study: not injected Tau− (zebrafish specimens with no Tau and no GNPDA2); not injected Tau+ (zebrafish specimens overexpressing human TauP301L and no GNPDA2); cLUC Tau− (zebrafish specimens with no Tau and expressing the luciferase vector); cLUC Tau− (zebrafish specimens overexpressing human TauP301L and expressing the luciferase vector); GNPDA2 Tau− (zebrafish specimens with no Tau and expressing human GNPDA2) and GNPDA2 Tau+ (zebrafish specimens overexpressing human TauP301L and expressing human GNPDA2).

**Figure 5 biomolecules-14-00394-f005:**
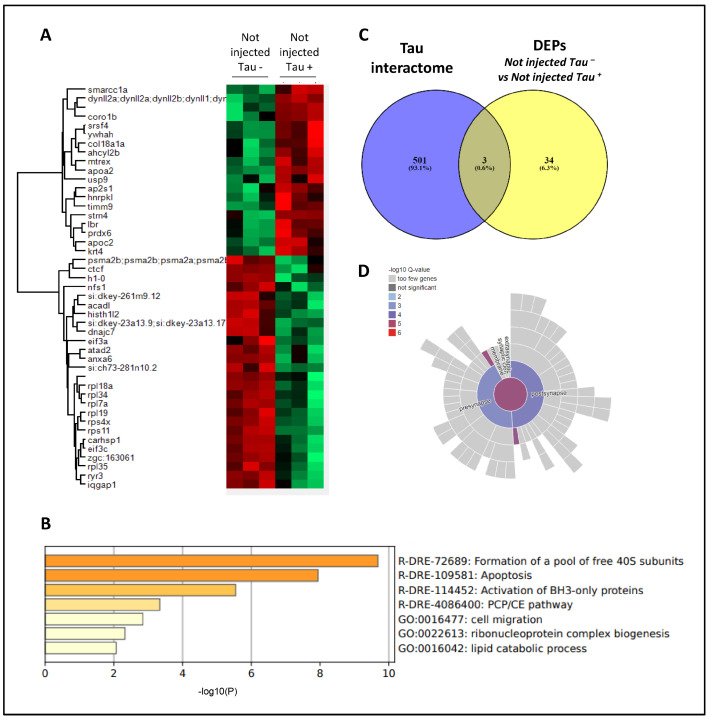
Proteomics in h.Tau P301L zebrafish embryos. (**A**) Heatmap representation showing differentially expressed proteins (DEPs) between not-injected Tau− and not-injected Tau+. (**B**) Functional clustering of DEPs between not-injected Tau– and not-injected Tau+ using Metascape. (**C**) Venn diagram showing DEPs previously described as Tau interactors. (**D**) Synaptic ontology analysis (subcellular distribution) of DEPs using the SynGo tool.

**Figure 6 biomolecules-14-00394-f006:**
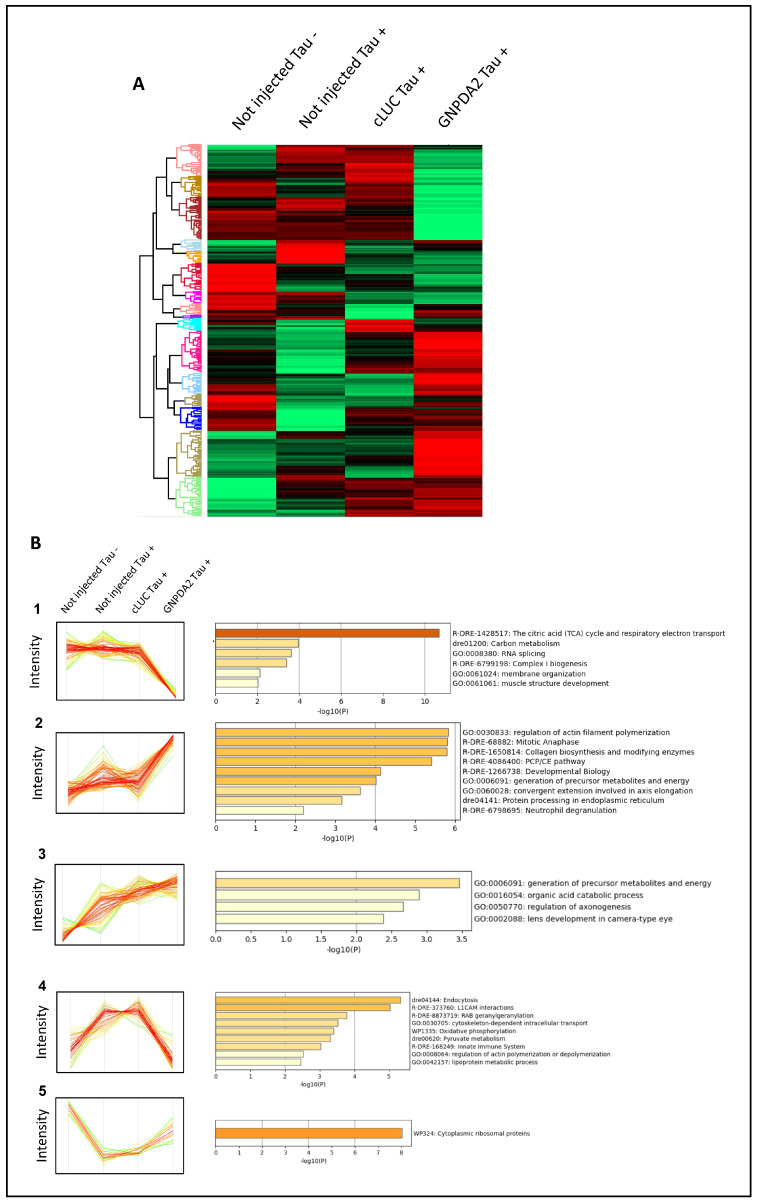
Differentially expressed proteins across Tau P301L embryos overexpressing GNPDA2. (**A**) Heatmap representing the differential zebrafish proteome across not-injected Tau−, not-injected Tau+ embryos, cLUC Tau+ and GNPDA2 Tau+ embryos. (**B**) Protein clusters specifically modulated across the different experimental conditions and functional clustering based on the specific disrupted zebrafish proteomes.

**Figure 7 biomolecules-14-00394-f007:**
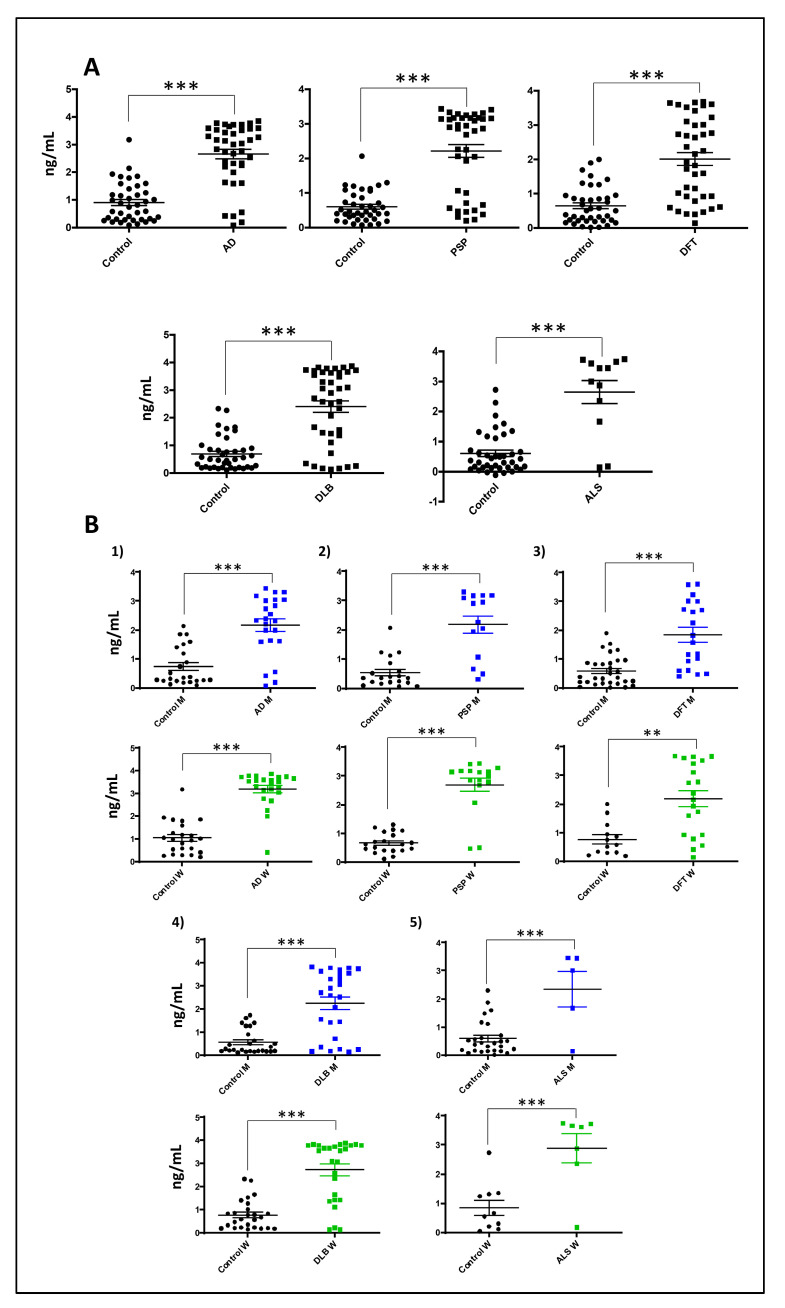
**Serum GNPDA2 levels across neurological disorders**. (**A**) GNPDA2 levels measured in the sera derived from 215 individuals with different neurological syndromes. (**B**) Sex-specific analysis across neurological disorders (71 controls; mean age: 69.2 years; 40M/31F, 40 AD subjects: mean age: 75.1 years; 20M/20F, 40 PSP subjects; mean age: 67.9 years; 15M/25F, 40 DFT subjects; mean age: 69.5 years; 20M/20F, 12 DLB subjects; mean age: 73.5 years; 6M/6F; 12 ALS subjects; mean age: 57.8 years; 5M/7F; ELISA (Mann–Whitney U test; ** *p*-value < 0.01; *** *p*-value: 0.001)).

**Table 1 biomolecules-14-00394-t001:** Samples included in GNPDA2 serum analysis.

Group	N° of Patients (Mean Age)	Male	Female
Control	71 (69.2)	40	31
Alzheimer’s disease	40 (75.1)	20	20
Progressive supranuclear palsy	40 (67.9)	15	25
Frontotemporal dementia	40 (69.5)	20	20
Dementia with Lewy bodies	12 (73.5)	6	6
Amyotrophic lateral sclerosis	12 (57.8)	5	7

## Data Availability

Mass spectrometry data and search results files were deposited in the ProteomeXchange Consortium via the JPOST partner repository (https://repository.jpostdb.org) with the identifier PXD049015 for ProteomeXchange and JPST002492 for jPOST (for reviewers: https://repository.jpostdb.org/preview/214159403965b93bb6e7293) (Accessed on 30 January 2024).

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
