# Peer review of "Involvement of Glucosamine 6 Phosphate Isomerase 2 (GNPDA2) Overproduction in β-Amyloid- and Tau P301L-Driven Pathomechanisms"

_biomolecules, 2024, doi:10.3390/biom14040394_

Round 1
Reviewer 1 Report
Comments and Suggestions for Authors
The manuscript by Mercedes Lachen-Montes and colleagues is an interesting study of GNPDA2 interactome in nasal epithelial cells (NEC) based on RNA-seq and quantitative proteomics. Authors report that many of the GNPDA2 interactors are involved in intraciliary transport. Overexpression of GNPDA2 in NEC was shown to promote proliferation. Interestingly, exogeneously supplied amyloid beta or overexpressed TauP301L lead to reduction in proliferation of NEC, the two targeting differentially at transcriptomic level--amyloid beta mostly affecting cytoskeletal pathways while TauP301L affecting membrane-mediated transport pathways. GNPDA2 overexperssion in zebra fish embryos, however, does not prevent taupathy caused by co-expresses TauP301L even though there was significant proteomics alterations in co-expressing embryos. Authors also found that GNPDA2 levels were higher in patient sera.
There are some concerns:
1) The major concern is with the experiments involving amyloid beta 1-42 peptide and TauP301L. These proteins may or may not be toxic depending on their aggregation state. It is not clear from the manuscript if these proteins are in monomeric or oligomeric or aggregated form at the time the sample was prepared for RNA-seq. Additionally, there is no evidence showing if exogeneously supplied amyloid beta 1-42 actually entered the cells.
2) Another concern is related to the contradictory proposition about levels of GNPDA2 in the manuscript. Line 7 of Abstract, “Moreover, GNPDA2 overexpression...protein processing” suggests protective effects of GNPDA2. However, lines 14 , “In humans, a significant ...(n=187)” and section 3.5 suggest pathogenic consequences of higher GNPDA2 levels. This is rather contradictory.
3) Why was Amyloid beta exogeneously supplied and not expressed like TauP301L in NEC?
4) Why was Amyloid beta not studied in zebra fish embryos even though amyloid beta showed significant effects in NEC.
Minor points:
“Not inject Tau-” and “Not inject Tau+” should be elaborated in main text too.
Please correct "Not inyect Tau-" and "Not inyect Tau+" in Fig 5A and Fig6 A, B.
The text and markers on all figures (particularly in Fig 3) need to be enlarged to be clearly visible.
There is significant verbatim text overlap in certain sections of the manuscript with a previous manuscript possibly from the same group. Please address this overlap.
References 6, 23, 45 do not have full citation.
Comments on the Quality of English Language
English needs minor editing.
Reviewer 2 Report
Comments and Suggestions for Authors
The Article by Mercedes Lachen-Montes et al is an interesting article one the involvement of GNPDA2 in Alzheimer and other brain diseases.
Concerning the science:
the authors are quite focussed on that Abeta and TAU are the cause of Alzheimer disease.
like the sentence in chapter 3.2
13,36], we explored the potential molecular events linked to a GNPDA2 overexpression in NECs in the presence of neuropathological insults (beta amyloid (Aβ) and mutated form of h.Tau P301L) what is the hen and what is the egg? in the articles opinionn it is Ab and Tau that come first and so GNPDA2 but it could well be the other way around?
Another question is if ERK is the right negative control for the immune prescipiotation studies. as is mentioned the expression of TAU P301L has an effect on ERK which indicates that ERK is involved in the effects.
Figure 3 is of not so much importance as it is almost impossible to see what is in the figure. maybe either make it a different way or take it out.
in figure 4 it is written:
These data indicate that overexpression of human GNPDA2 together with the h.Tau P301L mutation in zebrafish neurons does not modify the neuronal toxicity but slightly interferes with tau levels.
This is for me not visible.
In the discussion
Among numerous mechanisms that connect the neurotoxic Aβ and Tau are oxidation of proteins, lipid peroxidation and brain hypo-metabolism.
maybe make this statement more nuanced as has been shown clearing Ab does not have much of an effect on AD. furthermore, is AB and TAU the cause of the disease or is the disease the cause of AB and TAU?$
Interferon is shorted as IFN not INF
Comments on the Quality of English Language
In general the article is well written and good to follow. There are a few minor points in the English.
Like in the introduction
That is why, therapeutic strategies have been mostly Aβ and tau-targeted, however, any definitive suc-cess has been described until now. This should probably be: ...... No definitive success.....
Due to an increased in GNPDA2 protein levels have been previously
Due to an increased in GNPDA2 protein levels have which been previously ....
there are several of these small things where it would be good to look into.
Maybe it would be possible for the authors to make a cartoon in the discussion how they see the effect of GNDPA2 in the process of CNS disease.
Mitochondrial dysfunction seems to be more and more at the center of brain diseases and here GNDPA2 would fit perfectly in and could nuance the Ab and Tau influence.
Reviewer 3 Report
Comments and Suggestions for Authors
biomolecules-2894935. Manuscript review
The authors of this manuscript release an interesting study about the involvement of GNPDA2 protein levels and its activity in impairment or damage in neurodegenerative olfactory disorder models using nasal epithelial cells and transgenic zebrafish implicate in Alzheimer's disease hallmark, upon two main targets for this physiopathology as are App and TAU mutant P301L. Overall, the paper is well-written, and the experiments are technically great sound, and one light comment but are necessary to improve the manuscript.
Results:
Fig.1. The authors should indicate in the text and Fig. 1A that GNPDA2 recombinant protein has a 6xHis-tag, and add a brief comment describing why GNPDA2 has a different band-shift to compare with GNPDA2 immunoprecipitated. Also, is necessary to enclose the specific molecular weight of GNPDA2 in WB.
Fig.2. Can the authors explain why lines or curve cell proliferation in GNP A2 OD and GNP A2 + Ab have different behaviors when compared with other graphs, for 24 h after treatment?
Fig.3. Fantastic procurement approaches were released by authors, in this figure, but could they complete it, and add biochemical data regarding some western-blot or mRNA analysis levels to confirm whether there are IFN alpha pathways, AKT signaling, and survival kinases such as p38 or ERK? It would be very interesting to increase the manuscript value.
Fig.4. I want to ask the authors if they could be able to add additional analysis of pTAU levels. Probably, the specific epitope of pTAU Ser396 isn´t enough evaluate to the possibility that there are impairments or improvements when it is overexpressed GNPDA2. I propose they can use pTAU clon AT8 because is one great antibody for evaluating tauopathies.

Reviewer 4 Report
Comments and Suggestions for Authors
In this super complex manuscript the authors aimed to describe the role of GNPDA2 in the pathogenesis of AD. Please find my comments below:
· Introduction: no major issues detected. Authors clearly describe that olfaction loss is an early symptom of AD. As this has been associated with Aβ accumulation and hyperphosphorylated tau, therapies are Aβ and tau-targeted without any success. Based on the fact that decreased glucose level in the brain is the hallmark of AD, connection between glucose metabolism and AD is not well understood. Since previous reports described alterations in GNPDA2 expression in the olfactory bulb from AD and PD patients, the authors aimed to increase the knowledge about GNPDA2 in AD-related context.
· Materials and methods: authors applied a wide variety of methods in this study. Please review this part, since in some cases manufacturers are missing, AB concentrations applied are also missing. It would support a better understanding if basic patient data would be described in a table format instead of integrating into the text (section 2.13).
· Figures: small modifications are needed. In case of Figure 2B, it is hard to read numbers on the axes of the volcano plots, please modify. Figure 3: what does green and red represent? Please describe in the figure legend. Figure 5 and Figure 6, please check for typos.
· Results are written in an accurate and detailed way, however, regarding serum ELISA data (represented at Figure 7B) I do not agree with the conclusion that GNPDA2 expression level is gender-independent. Here, healthy and sick men and women are compared. If gender dependency is checked, comparison between men and women should be performed.
· Discussion: well written, conclusions are supported by the results. Based on RNA-Seq data, authors described novel pathways (i.e. axonogenesis, citric acid cycle, etc) in which GNPDA2 plays essential role.
Comments on the Quality of English LanguageIn this complex manuscript, authors aimed to clarify the olfactory nature of AD by investigating alterations in GNPDA2. For this, wide variety of models (including in vitro human cell lines, in vivo Zebrafish and patient-derived serum) and methods (mass spectrometry, immunoprecipitation, RNA-Seq, etc…) were used.
Sections of the manuscript are well balanced, no major issues were detected. Slight modifications are required in figures, materials and methods and at the conclusion of ELISA experiments. In case of serum ELISA experiments, authors say that the serum expression level is gender independent, however at Figure 7B healthy controls and patients are compared.
Regarding to English language, no major issues were detected, however, typos should be corrected.
After minor modifications, manuscript is acceptable for publication.
Round 2
Reviewer 1 Report
Comments and Suggestions for Authors
The revised version of the manuscript and the accompanying author's response adequately address most of the concerns raised in the previous round of the review.
However, the text and markers in all the figures are still difficult to read. Authors must improve the readability of the figures for the benefit of the potential readers of their research work.
Comments on the Quality of English LanguageEnglish need minor editing.